# Dynamic architecture of mycobacterial outer membranes revealed by all-atom simulations

Turner P Brown[1], Matthieu Chavent[2], Wonpil Im[1,3]*

[1]Department of Bioengineering, Lehigh University, Bethlehem, United States; [2]Université de Toulouse, CNRS, Laboratoire de Microbiologie et de Génétique Moléculaires, CBI, Toulouse, France; [3]Department of Biological Sciences, Lehigh University, Bethlehem, United States

## eLife Assessment

In their study, Brown et. al. provide an **important** advance in understanding the architecture of the mycobacterial outer membrane. Using all-atom simulations of model mycomembranes, the work reports **compelling** structural insights into how α-mycolic acids and outer leaflet lipids (PDIM and PAT) shape membrane organisation. The work revealed membrane heterogeneity with ordered inner leaflets and disordered outer leaflets that provide a molecular explanation for the resilience of the mycobacterial envelope.

*For correspondence: wonpil@lehigh.edu

Competing interest: The authors declare that no competing interests exist.

**Abstract** Tuberculosis remains a global health crisis due to the resilience of *Mycobacterium tuberculosis* (*Mtb*), largely attributed to its unique cell envelope. The impermeability and structural complexity of the outer membrane of this envelope, driven by mycolic acids and glycolipids, pose significant challenges for therapeutic intervention. Here, we present the first all-atom models of an *Mtb* outer membrane using molecular dynamics simulations. We demonstrate that α-mycolic acids adopt extended conformations to stabilize bilayers, with a phase transition near 338 K that underscores their thermal resilience. Lipids in the outer leaflet, such as PDIM and PAT, induce membrane heterogeneity, migrating to the interleaflet space and reducing lipid order. The simulated mycobacterial outer membrane has ordered inner leaflets and disordered outer leaflets, which contrasts with the outer membrane of Gram-negative bacteria. These findings reveal that PDIM- and PAT-driven lipid redistribution, reduced lipid order, and asymmetric fluidity gradients enable *Mtb's* outer membrane to resist host-derived stresses and limit antibiotic penetration, thereby promoting bacterial survival. Our work provides a foundational framework for targeting the mycobacterial outer membrane in future drug development.

## Introduction

Tuberculosis (TB) has been known to humankind since ancient times. Commonly known as Phthisis in ancient Greece and Consumption in the 1800s, TB has a storied history, influencing cultures around the world and being notoriously difficult to treat. Despite increasing investments into detection and treatment development, an estimated 1.25 million people died of TB in 2023. This makes it the world's deadliest infectious disease, surpassing COVID-19 (*Global Tuberculosis, 2024*). *Mycobacterium tuberculosis* (*Mtb*), the etiologic agent of TB, was first discovered in 1882 by Dr. Robert Koch. TB is a disease with a wide spectrum of manifestations – some individuals have active uncontrolled disease when the bacteria are actively replicating, whereas others may have controlled or latent infection

**Figure 1.** Schematic of a mycobacterial outer membrane composed of mycolic acids (MAs), trehalose di- and mono-mycolate (TDM and TMM), phthiocerol dimycocerosate (PDIM), sulphoglycolipids (SGL), and pentacyl and diacyl trehalose (PAT and DAT). This membrane is covalently attached to the peptidoglycan via the arabinogalactan.

when the bacteria may lie dormant intracellularly in macrophages or other cells. *Mtb* is a slow-growing bacterium with a formidable cell wall characterized by acid fastness. Antibiotics targeting the cell wall (Isoniazid, Pretomanid) are prominent in multi-drug regimens for drug-susceptible and drug-resistant strains. However, these regimens take many months, and drug-resistant *Mtb* strains are an increasing problem, necessitating further drug development (*Dartois and Dick, 2024*).

The cell envelope of *Mtb* is unusually thick and impermeable. The innermost layer is a plasma membrane consisting of various phosphatidyl-*myo*-inositol mannosides (PIMs) and lipomannan. Just above the plasma membrane is a peptidoglycan layer bound to a layer of arabinogalactan (*Figure 1*). Arabinogalactan sugars are covalently linked to mycolic acids in the inner leaflet of the mycobacterial outer membrane (MOM), also known as the mycomembrane. This is the bacterium's first defensive layer against external threats. The MOM is an asymmetric bilayer and the most complex layer of the cell envelope. Although its composition was unknown for decades, advancements in biochemical assays have yielded a reliable description of the native mycomembrane (*Chiaradia et al., 2017*). Lipoarabinomannan is also present in the cell envelope, but its exact location is up for debate. Many models show it anchored in the inner membrane, but it has been shown to be surface-exposed, suggesting that it extends through the periplasmic space and the outer membrane (*Mishra et al., 2011*; *Torrelles and Chatterjee, 2023*).

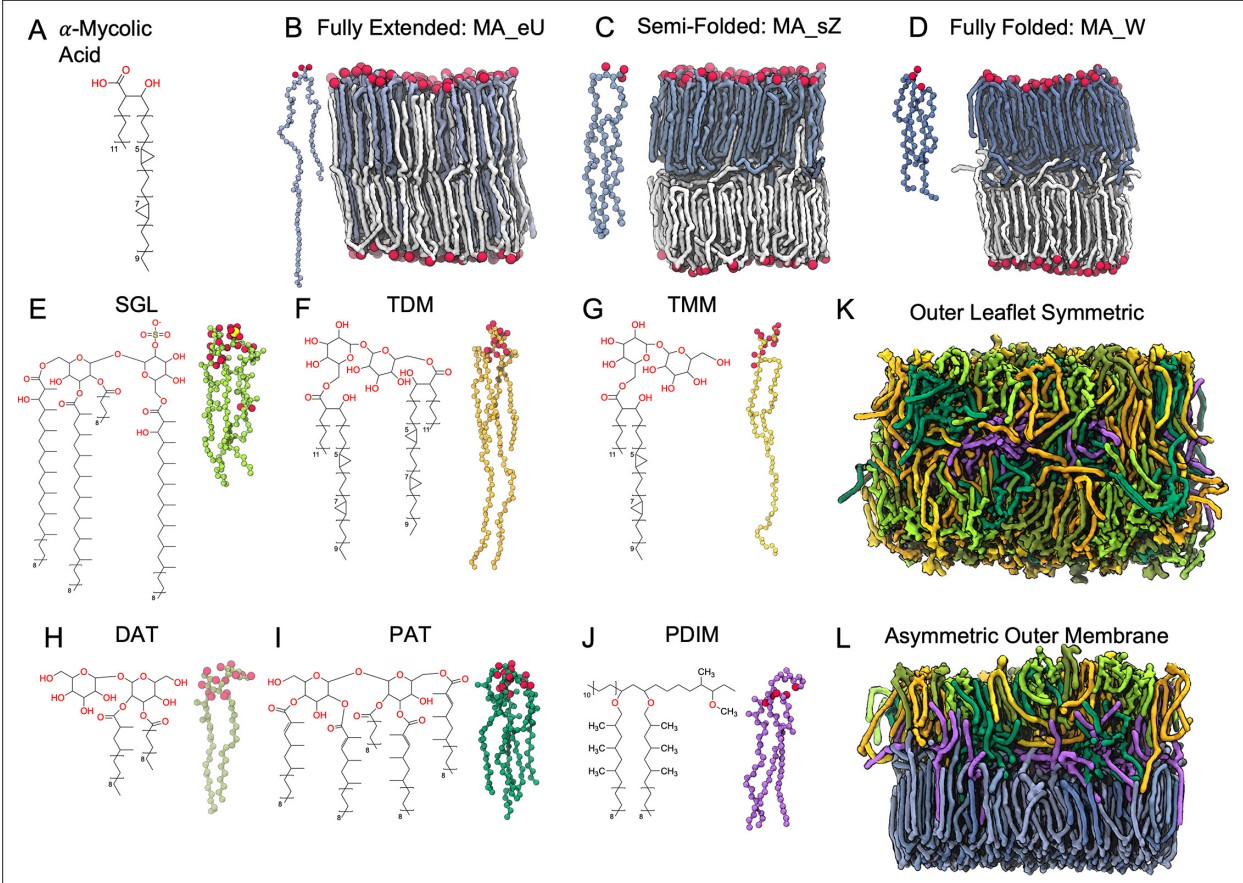

**Figure 2.** Chemical structures of mycobacterial lipids and membranes snapshots. (**A**) Chemical structure of α-mycolic acid (α-MA) with 78 carbons. (**B–D**). Initial three-dimensional (3D) conformations (left) and snapshots after 3 μs production of pure mycolic acid bilayers (right) with fully extended (MA_eU), semi-folded (MA_eZ), and fully folded (MA_W) conformations. Lipids in the upper leaflet are colored in light, medium, and dark blue, and lipids in the lower leaflet are colored in white. Oxygen atoms are shown as van der Waals spheres. (**E–J**) Chemical structures (left) and initial conformations (right) of the six outer leaflet lipids. (**K**) Snapshot after 3 μs production of the symmetric system containing outer leaflet lipids at 313 K (All_Lipids_313). (**L**) Snapshot after 3 μs production of the whole mycobacterial outer membrane with asymmetry of lipids at 313 K (Asym_313 system). Lipid colors in (**K**) and (**L**) match the colors of the initial 3D conformations.

The mycomembrane inner leaflet is composed of ultra-long-chain fatty acids known as mycolic acids (MAs). MAs, which contain between 60 and 90 carbon atoms, consist of a long β-hydroxy chain and a short α-alkyl side chain (*Figure 2A*). Further classification of MAs is based on functional groups, categorizing them into three groups: alpha, methoxy, and keto-MA. α-MAs contain cyclopropane groups at two positions along the β-hydroxy chain (proximal and distal); methoxy-MAs contain a methoxy group in the distal position; and keto-MAs contain a carbonyl group at the distal position (*Marrakchi et al., 2014*). These three MAs exist in varying concentrations depending on the species, and were shown to have distinct effects on monolayer and bilayer characteristics (*Hasegawa and Leblanc, 2003*; *Kumar et al., 2025*). α-MAs are of particular interest not only because of their extreme length, but also the presence of cyclopropane rings (*Figure 2A*). Such ultra-long-chain fatty acids are able to adopt various folding patterns, which helps to regulate bilayer order and symmetry (*Kawaguchi et al., 2022*). Cyclopropanation of acyl chains tends to promote bilayer fluidity by interfering with lipid packing, enhancing the formation of *gauche* defects, and increasing lipid diffusion. Based on bilayer simulations, Poger et al. showed that fatty acid cyclopropanation increases fluidity in the plasma membrane. Additionally, they found that *cis*- and *trans*-cyclopropane fatty acids have distinct ordering effects (*Poger and Mark, 2015*). In the case of MAs, the first observation of distinct folding patterns was made by Takeshi Hasegawa in 2004 using surface-enhanced Raman scattering (*Hasegawa, 2004*). Later studies described three major folding patterns in MAs (*Groenewald et al., 2019*; *Groenewald et al., 2014*; *Savintseva et al., 2023*): W, a fully folded shape in which the long

chain folds in on itself twice (four parallel chains); Z, a semi-folded shape in which the long chain folds in on itself once (three parallel chains); and U, a fully extended shape in which the long chain does not fold in on itself (two parallel chains; *Figure 2B–D*). Although these conformations were observed in experimental and computational studies, they do not represent the complete diversity of folds that MAs can exhibit. In this study, we have analyzed the folding behavior and effects on bilayer stability of α-MAs in a mycomembrane-like environment. We have also used the equilibrated symmetric bilayers to estimate reasonable areas per lipid and facilitate the modeling of stable asymmetric systems.

The outer leaflet of the mycomembrane contains many structurally and functionally diverse glyco-lipids, which aid in pathogenicity and virulence of *Mtb*. Phthiocerol dimycocerosate (PDIM, *Figure 2J*) is a long-chain, nonpolar lipid found on the surface of *Mtb* and other pathogenic slow-growing myco-bacteria (*Daffe and Laneelle, 1988*). It consists of a phthiocerol A backbone, connected to two methyl-branched fatty acids by ester linkages. Loss of PDIM is associated with decreased virulence (*Mulholland et al., 2024*). Additionally, PDIM contributes to pathogenesis by promoting escape from phagolysosomes (*Augenstreich et al., 2017*; *Barczak et al., 2017*; *Lerner et al., 2018*; *Quigley et al., 2017*). Trehalose dimycolate (TDM, *Figure 2F*), also known as cord factor, and trehalose mono-mycolate (TMM, *Figure 2G*) are the most abundant glycolipids in *Mtb*. TDM consists of a trehalose head group connected to two MA tails through ester linkages, whereas TMM only contains one MA tail. TDM promotes *Mtb* survival by decreasing phagosomal acidification and phagolysosomal fusion in macrophages (*Indrigo et al., 2003*), whereas strains lacking TDM exhibit reduced infectivity in vivo and survival in vitro (*Bloch, 1950*; *Indrigo et al., 2002*). Diacyl trehalose (DAT, *Figure 2H*) consists of a trehalose headgroup with one steric acid tail and one mycosanoic acid tail. Pentacyl trehalose (PAT, *Figure 2I*) consists of a trehalose headgroup with one steric acid tail and four mycolipenic acid tails. Sulfoglycolipid (SGL, *Figure 2E*) consists of a sulfated trehalose headgroup with one palmitic acid tail, one phthioceranic acid tail, and two hydroxyphthioceranic acid tails. These lipid tails contain a varying number of methyl groups close to the headgroups. For decades, significant efforts have been made to elucidate the biosynthetic pathways of the trehalose-derived glycolipids (*Kalscheuer and Koliwer-Brandl, 2014*). Now that the composition of the mycomembrane is mostly known, an emerging ques-tion in the field is how this uniquely complex membrane dynamically protects the bacterium.

In lieu of expensive laboratory experiments restricted by biosafety considerations, researchers are increasingly adding computational simulations to their repertoires for studying biochemical systems. Molecular dynamics (MD) simulations employ Newton's equations of motion, along with descriptions of electrostatic and chemical interactions to simulate the motion of individual atoms with discrete time steps. Since the first simple MD simulations in the late 1950s, intense scientific efforts and advance-ments in computational power have increased the accuracy and size of these simulations manyfold. In recent years, a variety of full bacterial cell envelopes have been successfully simulated, such as the AcrAB-TolC multidrug efflux pump embedded in the *Escherichia coli* cell envelope. Specifically, much effort has been devoted to modeling the cell envelope of Gram-negative bacteria (*Khalid et al., 2022*). Recent work has expanded to mycobacterial systems. For instance, transport mechanics of the Mycobacterial membrane protein large 3 (MmpL3) have been modeled via high-throughput virtual screening and MD simulations (*Choksi et al., 2024*). Additionally, the dynamic behavior of MAs in a variety of environments has been simulated, connecting their conformations and functional groups to differences in drug permeability (*Basu et al., 2024*; *Groenewald et al., 2014*; *Vasyankin et al., 2024*). The supramolecular organization of PIM lipids in the mycobacterial plasma membrane has been mapped, showing how these lipids cluster to modulate membrane fluidity and host-pathogen interactions (*Brown et al., 2023b*; *Brown et al., 2023a*). Finally, simulations of PDIM lipids demon-strated their conical shape promotes macrophage membrane remodeling, a key step in *Mtb* infection (*Augenstreich et al., 2019*).

Despite these advances, modeling the mycomembrane with MD simulations remains uniquely chal-lenging due to its dense array of structurally diverse components, its complex spatial organization, and the paucity of resolved protein structures. To overcome these limitations and advance atomistic modeling of the mycobacterial envelope, we employed a three-step approach in this study. First, symmetric bilayers consisting of α-MAs in various starting conformations were simulated to elucidate whether they would form stable bilayers and to observe the various folding patterns (*Figure 2B–D*). Next, symmetric bilayers with various *Mtb*-specific glycolipids were simulated (*Figure 2K*). Finally, the area-per-lipid (APL) values from the inner leaflet symmetric bilayers were used to construct asymmetric

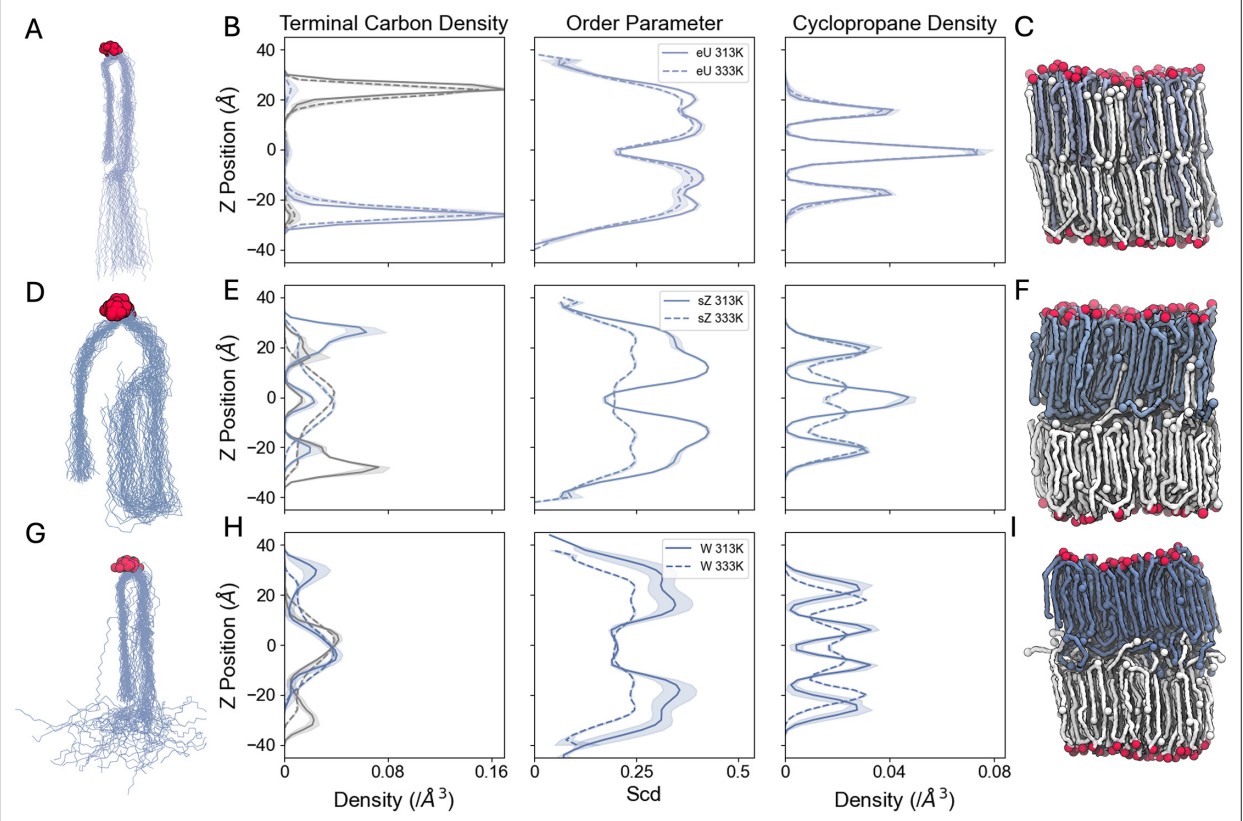

**Figure 3.** Folding patterns and organization of mycomembrane inner leaflet symmetric bilayers. (A, D, G) Overlaid structures of one mycolic acid (A MA_eU, B MA_eZ, and C MA_W) from every 10 frames of the final 500 ns production. Structures were aligned on 15 carbons at the top of the α-alkyl chain and 3 carbons at the top of the β-hydroxy chain: named C10 to C27 in the MA topologies. Oxygen atoms are displayed as red spheres. (B, E, H) Lipid dynamics at 313 K (solid lines) and 333 K (dashed lines). The three columns are terminal carbon density profiles, average lipid order parameters of carbons (*Vermeer et al., 2007*), and cyclopropane density profiles in 2 Å-wide slabs along the z-axis (i.e. the membrane normal) with the bilayer center at z=0. For carbon density profiles, lipids belonging to the lower leaflet are depicted in grey and those from the upper leaflet are depicted in blue. (C, F, I) Snapshots with varying initial compositions after 3 μs production. Lower leaflet lipids are shown in white, upper leaflet lipids are shown in light, medium, and dark blue. Oxygen atoms are displayed as red spheres. Terminal carbons and cyclopropane carbons are shown as spheres colored according to their leaflet.

The online version of this article includes the following figure supplement(s) for figure 3:

**Figure supplement 1.** Time series of inner leaflet symmetric bilayers with various starting conformations of α-mycolic acid.

MOM bilayers with α-MAs in the inner leaflet, and six lipid types in the upper leaflet (*Figure 2L*). The main goals of this study are to characterize the dynamics of mycobacterial lipids in realistic MOM bilayers and to elucidate the organizational structure of the mycomembrane. To the best of our knowledge, this study presents the first full atomistic models of the mycomembrane, an important step towards modeling the full mycobacterial cell envelope.

## Results
### α-Mycolic acid conformations dictate stability and thermal resilience in symmetric bilayers

The first set of simulations were run to characterize the stability of three conformations of α-MAs. We simulated three distinct bilayers, with MAs initially in fully extended (MA_eU), semi-folded (MA_sZ), or fully folded (MA_W) conformations (*Figures 1 and 2*). These compositions were simulated at 313 K and 333 K to investigate the stability of α-MAs at high fever temperature. The simulation system name, compositions, and basic statistics are summarized in *Supplementary file 1*.

From our simulations, MA_eU (*Figure 3A*) and MA_sZ (*Figure 3D*) mostly maintained their initial conformations in the bilayers, while MA_W quickly unfolded into the sZ shape (*Figure 3G*) by extending their long chains toward the midplane of the bilayer, encountering a barrier to interdigitation, and folding back into their own leaflet. This causes an increase in the membrane thickness (*Figure 3—figure supplement 1A*). As shown in *Figure 3E and H*, MA_sZ and MA_W have wider terminal carbon distributions than MA_eU, indicating a variety of conformations. When compared to the profiles from the MA_eU systems (*Figure 3B*), there is significantly less interdigitation by the long chains of MA_sZ and MA_W. We believe that the barrier to interdigitation at the interleaflet space is due to L-shaped conformations, in which α-MA long chains first extend parallel to the bilayer normal, then undergo a sharp kink at the bilayer center, causing the carbons beyond the first cyclopropane ring to settle perpendicular to the bilayer normal. This leads to vertical heterogeneity in the bilayers with initially folded MAs. In *Figure 3B, E and F*, drops in the order parameter profiles align well with the density of cyclopropane rings, indicating a bilayer disrupting effect. These trends are visually displayed in *Figure 3C, F and I*, where snapshots of the three bilayer systems at 313 K are shown. In the MA_W snapshot (*Figure 3I*), the terminal carbons of lipids from each leaflet are mixing at the interleaflet space, and do not interdigitate as much as in the MA_eU or MA_sZ systems. These results indicate that fully folded α-MAs cannot form a stable bilayer and prefer to be in extended conformations. This is substantiated by the time series of the membrane thickness (*Figure 3—figure supplement 1A*), in which the MA_W bilayers do not reach a stable thickness even after 3 µs production. In the MA_eU systems, very few lipids transitioned to the sZ shape. In contrast, lipids in the MA_sZ system were able to transition to the eU shape. This asymmetry in conformational interconversion, in which sZ transitions to eU, but not vice versa, may reflect both thermodynamic preferences (e.g. lower energy barriers for sZ to eU) and the limited time scales of our simulations (3 µs production runs). While the MA_W system failed to equilibrate fully (*Figure 3—figure supplement 1A*), the lack of eU to sZ transitions in other systems could stem from kinetic trapping, where higher energy barriers or slower reorientation dynamics prevent sampling of these conversions within the simulation window.

To examine the temperature-dependence of α-MA folding, we simulated a symmetric bilayer of 60 MA_eU in each leaflet for 2 µs at 313, 323, 333, 338, 343, and 353 K. As temperature increases,

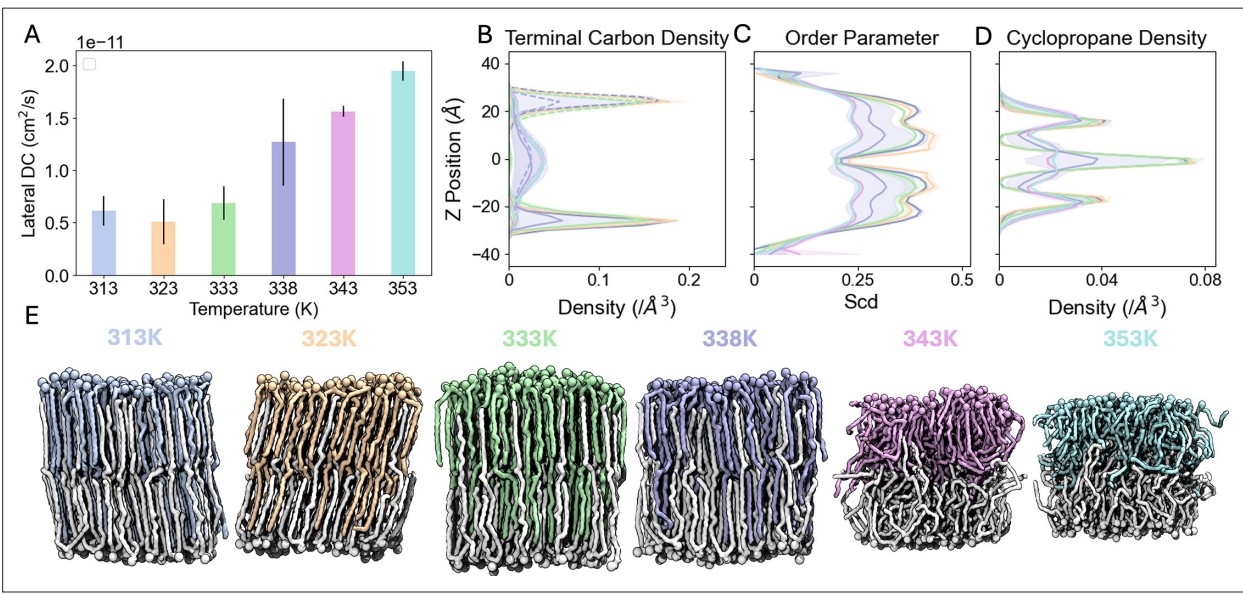

**Figure 4.** Phase transition of pure fully-extended α-mycolic acid bilayers. (**A**) Lateral diffusion coefficients from 313 K to 353 K. (**B**) Terminal carbon (C75) z-density profiles separated by leaflet along the temperature gradient. Upper leaflet lipids are shown with solid lines, and lower leaflet lipids shown with dashed lines. (**C**) Average order parameters of MA at varying temperatures. (**D**) Cyclopropane carbons (CC1, CC2) z-density profiles along the temperature gradient. (**E**) Snapshots along the temperature gradient, with lower leaflet lipids shown in white, upper leaflet lipids shown in colored lines, and oxygen atoms shown as spheres. Line and shaded area colors in (B, C, and D) correspond to those of upper leaflet lipids in **E**.

The online version of this article includes the following figure supplement(s) for figure 4:

**Figure supplement 1.** Time series of MA_eU bilayers at increasing temperature.

the average area per lipid (APL) increases linearly, while the membrane thickness decreases slightly (*Figure 4—figure supplement 1*). From 323 K to 353 K, the average lateral diffusion coefficients of MAs increase linearly, showing faster dynamics at higher temperatures (*Figure 4A*). The order parameter profiles of the long chain of MA show consistent drops in acyl chain order at the *proximal* and *distal* positions of the cyclopropane rings (*Figures 3C and 4B*), consistent with the MA_sZ and MA_W bilayers. As the system temperature increases from 313 K to 333 K, the average order decreases slightly. At 338 K, the lipid order drops significantly, and there is a high standard deviation among replicas. The 343 K and 353 K bilayers show a significantly lower order between the cyclopropane rings than the lower temperature bilayers, suggesting a phase transition from liquid ordered to liquid disordered (*Figure 4B*). Therefore, the transition temperature of bilayers with this composition appears to be close to 338 K in our simulations. Next, to understand how chain interdigitation changes as temperature increases, we produced z-density profiles of the terminal carbons (C75) from MAs and separated them by leaflet. As shown in *Figure 4D*, bilayers from 313 K to 333 K show a consistent distribution, with all lipids reaching far into the opposite leaflet. At 338 K, the density plot has an intermediate distribution between the low and high temp simulated bilayers, indicating that 338 K may be close to the melting point of the bilayer. The 343 K and 353 K profiles (pink and cyan lines, respectively) indicate that a phase transition has occurred. The carbons below the second cyclopropane rings in each leaflet mix in the interleaflet space, inducing a melted phase. In these higher temperature systems, α-MAs are very dynamic, similar to lipids in a liquid disordered phase. Having established α-MAs' preference for extended conformations and contributions to bilayer stability, we next explored how outer leaflet lipids fine-tune membrane properties under shifting metabolic and environmental pressures.

## Outer leaflet lipids drive unexpected membrane heterogeneity and softness of the mycomembrane

14 symmetric bilayer systems were simulated to investigate the effects of glycolipids in the mycomembrane outer leaflet (*Supplementary file 1*). The simulations have seven distinct compositions: one with all six glycolipids (*Figure 2E–J*), and the other six with one glycolipid omitted, mimicking specific lipid knockout strains. Each lipid type contains a trehalose headgroup initially at the water-membrane interface and a variety of acyl or MA tails. PDIM, the only lipid type without a trehalose headgroup, is a nonpolar molecule, and has been shown to migrate into host epithelial membranes, promoting infectivity (*Cambier et al., 2020*). In a 2019 study, PDIM lipids were shown to adopt a conical shape, allowing them to aggregate in between the two leaflets of a POPC bilayer (*Augenstreich et al., 2019*). In our simulations, PDIM moves quickly into the membrane center, creating a distinct layer at the interleaflet space (*Figure 5B and D*) and mimicking behavior from the previous study (*Augenstreich et al., 2019*). Surprisingly, some PAT lipid headgroups were also able to move into the bilayer center (*Figure 5B and D*). PAT has a hydrophilic trehalose headgroup, so their migration into the membrane center was unexpected. Notably, PAT has five lipid chains, which is the most chains out of all the glycolipids included in the systems. Also, methylation near the top of the lipid tails may increase the equilibrium APL of PAT and promote movement away from the water-bilayer interface into the membrane. Although PAT also migrates to the bilayer midplane, the PAT-deficient bilayers did not exhibit reduced thickness as the PDIM-deficient thickness did (*Supplementary file 1*). This may be due to fewer PAT than PDIM moving to the bilayer midplane. In the All_Lipids systems, PDIM migrates first, bulging the upper leaflet and reducing lipid headgroup crowding (*Figure 5—figure supplements 1 and 2*). In this slightly less crowded environment, hydrophobic forces from PAT's tails overcome the hydrophilic forces from the trehalose headgroup, causing some PATs to move deeper into the hydrophobic region. We also observed a stark difference in the speed with which PDIM and PAT migrate to the center at different temperatures. PDIM molecules do not fully aggregate at the membrane center until about 1500 ns at 313 K, whereas they accumulate within 500 ns at 333 K (*Figure 5B and D*). This can be attributed to higher kinetics at 333 K, causing the lipids to move faster. Coarse-grained models may be sufficient to observe full aggregation of hydrophobic species at the membrane midplane at lower temperatures. In the bilayers with no PDIM, less PAT migrates to the center, suggesting a link between induced disorder by PDIM and PAT migration (*Figure 5—figure supplements 1 and 2*). In all symmetric outer leaflet simulations, PDIM and PAT sit just below the headgroups of other lipids at the start of production, due to our equilibration scheme. During the last step of equilibration, lipid

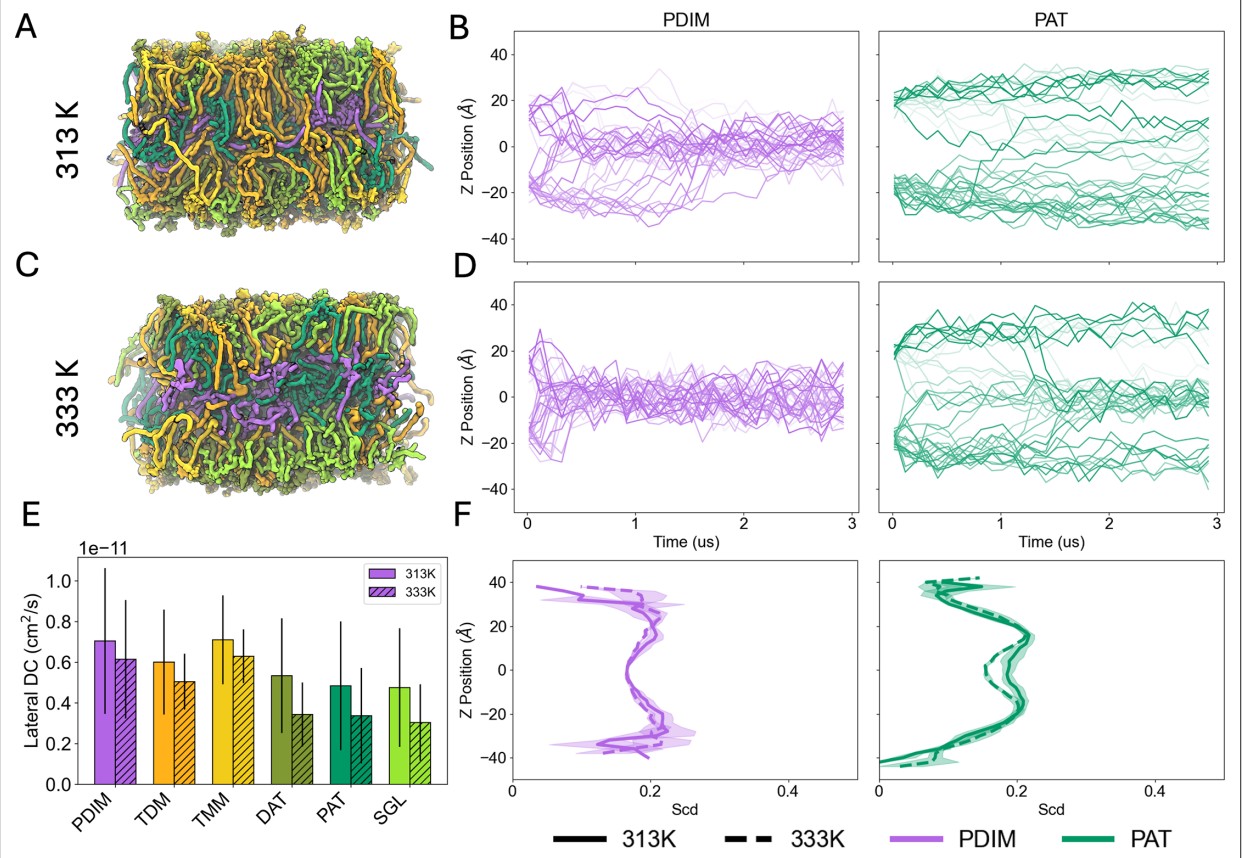

**Figure 5.** Lipid aggregation in mycomembrane outer leaflet symmetric bilayers. (**A, C**) Snapshots of All_Lipids systems after 3 µs production at 313 K and 333 K, respectively (see ***Supplementary file 1*** for system name, composition, and basic statistics). Lipid colors match the coloring from ***Figure 2***. Water and ions are omitted for clarity. (**B, D**) Headgroup z position time series of PDIM (purple) and PAT (green) at 313 K and 333 K, respectively. Each line is a separate lipid molecule. (**E**) Lateral diffusion coefficients from All_Lipids system for each lipid type. Solid bars are from the 313 K system, and hatched bars are from the 333 K system. Error bars are the standard errors across three replicas. (**F**) Comparison of average deuterium order parameters of carbons along the z-axis for PDIM (light purple) and PAT (light green) at 313 K (solid lines) and 333 K (dashed lines). Shaded regions are the standard errors across three replicas. All analysis is averaged over the last 500 ns production.

The online version of this article includes the following figure supplement(s) for figure 5:

**Figure supplement 1.** Headgroup Z position time series for all lipid types and all outer leaflet symmetric systems at 313 K.

**Figure supplement 2.** Headgroup z position time series for all lipid types and all outer leaflet symmetric systems at 333 K.

**Figure supplement 3.** Outer leaflet lipid dynamics.

**Figure supplement 4.** Scd vs carbon number for outer leaflet symmetric bilayers.

**Figure supplement 5.** Time series of outer leaflet symmetric systems at 313 K and 333 K.

headgroups are allowed to move freely, which initiates migration to the membrane center and causes the slight difference between PDIM/PAT and the other lipids' headgroup positions (***Figure 5—figure supplements 1 and 2***). To observe how this migration affects the fluidity of these lipids, we plotted the average order parameters of carbon atoms along the z-axis, and found low lipid order, with a slight valley at +/-10 Å from the bilayer center (***Figure 5F***). In fact, there was little to no difference in lipid order between 313 K and 333 K in any lipid type (***Figure 5—figure supplement 3E***). Additionally, in SGL-deficient bilayers, fewer PDIMs and PATs move to the bilayer midplane. This may be due to the highly methylated lipid tails of SGL. When present in the bilayer, these methyl groups may disrupt lipid packing and increase fluidity, allowing more PDIMs to move into the hydrophobic region. ***Figure 5—figure supplement 4*** shows the average lipid order parameter along each lipid tail for all outer leaflet symmetric systems. Without SGL, lipid tails are consistently more ordered, supporting the notion that SGL's methylated tails are disrupting lipid packing. Further studies are necessary to investigate the effect of glycolipid-deficient compositions on the dynamic properties of the asymmetric MOM.

The migration of PDIM and PAT into the interleaflet space disrupts lipid packing in the outer leaflet, creating localized regions of structural disorder. While these perturbations modestly elevate lateral diffusion coefficients compared to tightly packed regions, a signature of reduced packing efficiency, the absolute diffusion values remain strikingly low relative to typical lipid bilayers (*Figure 5E*; *Almeida et al., 2005*). This suggests that while lipid mobility increases at disordered sites, lateral movement across the membrane remains constrained, likely due to persistent interactions between long-chain mycolic acids or residual ordered domains. To bridge insights from symmetric bilayer simulations with the native mycomembrane, we constructed asymmetric bilayers replicating its architectural complexity.

## Asymmetric mycomembrane presents a phase transition from disordered outer leaflet to ordered inner leaflet

Simulating asymmetric bilayers presents a fundamental challenge: the intrinsic APL mismatch between leaflets with distinct lipid compositions. In asymmetric systems, differential packing of lipids in each leaflet can lead to bilayer instability, curvature, or defects if initial APLs are not carefully chosen

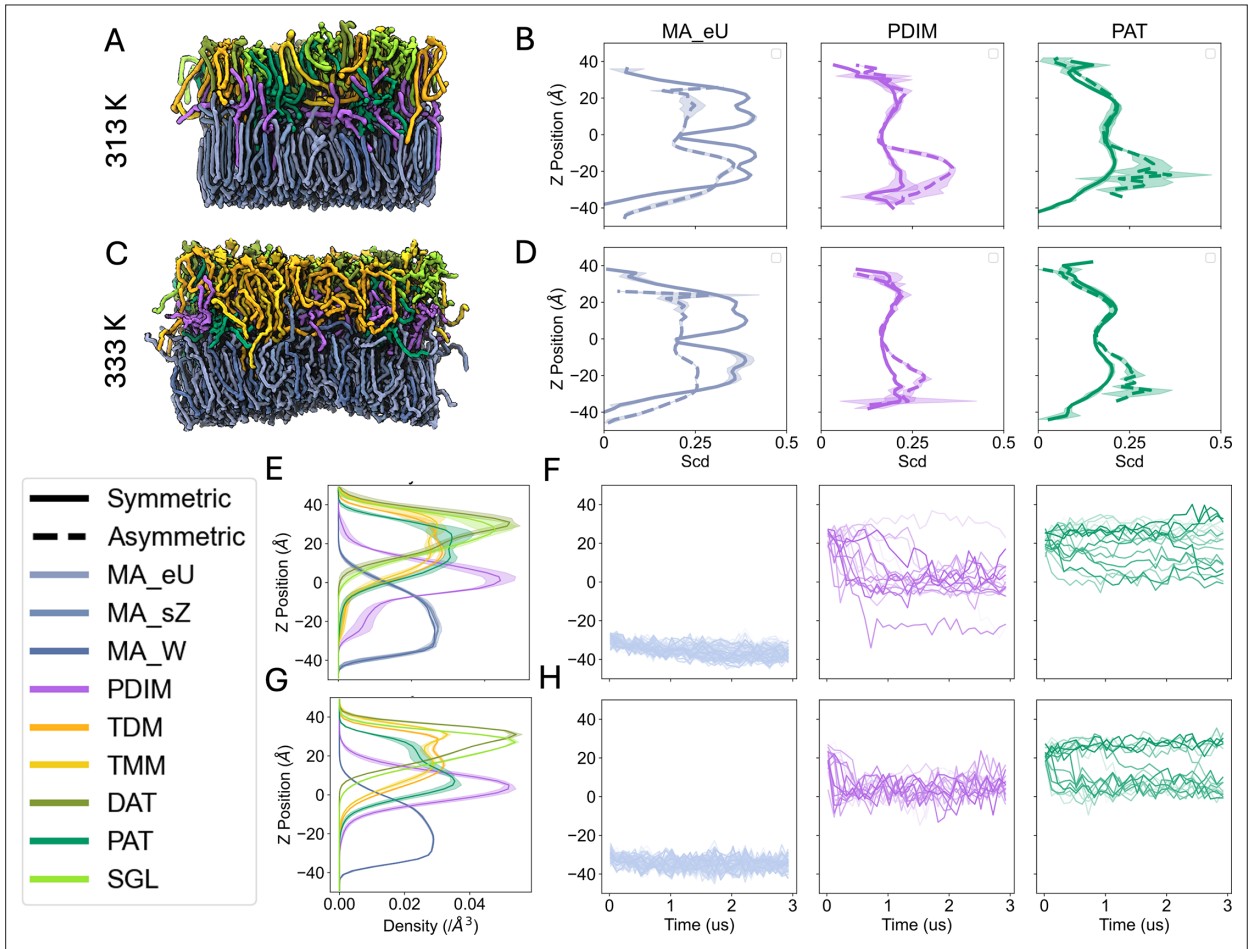

**Figure 6.** Vertical heterogeneity and lipid dynamics in asymmetric mycomembranes. (**A, C**) Snapshots after 3 μs production at 313 K and 333 K, respectively. Lipid colors match the coloring from *Figure 2*, and lipids are shown with the QuickSurf drawing method. (**B, D**) Comparison of average deuterium order parameters of carbons along the z-axis for MA_eU (light blue), PDIM (light purple), and PAT (light green) at 313 K and 333 K, respectively. Solid lines are from the symmetric systems and dashed lines are from the asymmetric systems. (**E, G**) Full lipid z-density profiles for each lipid at 313 K and 333 K, respectively. Shaded regions are the standard errors across three replicas. (**F, H**) Headgroup z position time series of MA_eU (light blue), PDIM (light purple), and PAT (light green) at 313 K and 333 K, respectively. Each line is a separate lipid molecule. All analyses are averaged over the last 500 ns of production.

The online version of this article includes the following figure supplement(s) for figure 6:

**Figure supplement 1.** Scd and headgroup time series for asymmetric systems.

(*Bodosa et al., 2024*; *Park et al., 2021*). For the mycomembrane, this challenge is compounded by the dynamic behavior of outer-leaflet glycolipids, such as PDIM and PAT, which migrate into the interleaflet space and disrupt conventional APL calculations (*Figure 5—figure supplements 1 and 2*). Specifically, Voronoi tessellation, a common method for estimating APLs, fails in these systems due to irregular spatial distribution and overlapping lipid tails. To address this, we calculated APLs from stable symmetric bilayers of α-MAs and adjusted the total lipid count in the asymmetric system to match the equilibrium APL of the outer-leaflet glycolipid ensemble (All_Lipids_313, see *Supplementary file 1*). The final asymmetric bilayer comprised 94 MA_eU, 47 MA_sZ, and 47 MA_W lipids in the lower leaflet, with 20 molecules of each glycolipid type in the upper leaflet. Although this model does not perfectly reflect a physiological composition of the mycomembrane as native mycomembranes have more TDM than TMM and PDIM, as well as a variety of MA classes in the inner leaflet, we believe that this composition is a good starting point, leading a way to more accurate models. This configuration was simulated for 3 μs at both 313 K and 333 K (*Figure 6A and C*), followed by analysis of temperature-dependent lipid dynamics.

We first calculated the average order parameter along the z-axis. When comparing the symmetric systems to the asymmetric system, we see a slight decrease in the order parameters for inner leaflet lipids at 313 K and a sharp decrease at 333 K (*Figure 6B and D*). Interestingly, PDIM and PAT, which also migrate to the bilayer center in the asymmetric systems, exhibit higher order in the inner leaflet. In fact, TDM and TMM also adopt a higher order when they reach into the inner leaflet (*Figure 6—figure supplement 1C and D*). Full lipid z-density profiles (*Figure 6E and G*), along with headgroup z-position time series, show that most of the PDIM molecules are settling in the interleaflet space (*Figure 6F and H*). Closer analysis reveals that PDIM lipids can fully flip-flop into the MA-rich inner leaflet, with some headgroups lining up with the upper bounds of the MA_eU headgroups. This suggests that, at thermodynamic equilibrium, the outer membrane of *Mtb* maintains heterogeneous fluidity, and PDIM migration to the membrane center is disrupting the deep interdigitation that stabilizes pure α-MA bilayers, allowing for a disordered outer leaflet. This unique membrane architecture has important implications for the resilience of *Mtb* in various growth states and host environments, which is elaborated below.

## Discussion

Our simulations of pure α-MA bilayers have revealed important dynamic behaviors that give evidence for resistance to passive diffusion and high thermal resilience. In contrast to common phospholipids, which have much shorter tails, α-MAs have a higher transition temperature. Such longer lipid tails have more surface area, increasing the strength of van der Waals interactions and raising their transition temperatures (*Rawicz et al., 2000*). This trend is consistent with our results. Additionally, the high degree of interdigitation prevents free lateral diffusion, further raising the melting temperature. Above 338 K, the degree of interdigitation drops and fluidity increases. The extended conformation of α-MAs and their phase transition near 338 K (~65 °C) in our simulations reflect a nuanced adaptation in *Mtb*. While this transition temperature exceeds host fever ranges (310–315 K), its biological significance may lie less in fever resistance and more in mitigating environmental stresses. Notably, the rigidity of MA-rich bilayers likely has evolved to withstand desiccation, critical for survival in aerosols during transmission, rather than extreme heat. This ordered structure creates a formidable barrier against antibiotics and host defenses, but its functionality extends further; under stasis conditions (e.g. hypoxia, caseum), *Mtb* downregulates trehalose dimycolate (TDM) production, potentially mimicking a membrane with fewer surface-exposed glycolipids (*Hunter et al., 2006*). Such metabolic adaptations could reduce fluidity, favoring a liquid ordered phase that further restricts permeability and sustains dormancy. A static, overly rigid membrane might fracture under stress, while excessive fluidity would permit drug entry. Based on our simulations, cyclopropane groups in MAs may balance these extremes, enabling structural flexibility without compromising impermeability. This 'dynamic resilience' explains *Mtb*'s resistance to small molecules; drugs targeting fluid membranes fail against the mycomembrane's adaptive plasticity (*Modak et al., 2022*). Importantly, the mycomembrane's adaptability underscores its vulnerability. Therapies disrupting MA cyclopropanation or folding dynamics could destabilize this equilibrium, collapsing the barrier.

The outer leaflet symmetric bilayers, comprised of trehalose-derived glycolipids and PDIMs, reveal PDIM-dependent thickness. As observed in both symmetric outer leaflet systems and asymmetric

systems, PDIM migrates to the bilayer midplane, causing the upper leaflet to bulge and increasing the overall thickness. Reduced thickness in the systems lacking PDIM, an important virulence factor for *Mtb*, may allow for higher nutrient uptake. This corroborates a 2009 study in which Domenech and Reed found a correlation between PDIM absence in vitro and attenuated virulence (*Domenech and Reed, 2009*). Although PAT also migrates to the bilayer midplane, the PAT-deficient bilayers did not exhibit reduced thickness as the PDIM-deficient thickness did. This may be due to fewer PAT than PDIM moving to the bilayer midplane. In the All_Lipids systems, PDIM migrates first, bulging the upper leaflet and reducing lipid headgroup crowding. In this slightly less crowded environment, hydrophobic forces from PAT's tails overcome the hydrophilic forces from the trehalose headgroup, causing some PATs to move deeper into the hydrophobic region. Additionally, in SGL-deficient bilayers, fewer PDIMs and PATs move to the bilayer midplane. This may be due to the highly methylated lipid tails of SGL. When present in the bilayer, these methyl groups may disrupt lipid packing and increase fluidity, allowing more PDIMs to move into the hydrophobic region. *Figure 5—figure supplement 4* shows the average lipid order parameter along each lipid tail for all outer leaflet symmetric systems. Without SGL, lipid tails are consistently more ordered, supporting the notion that SGL's methylated tails are disrupting lipid packing. Further studies are necessary to investigate the effect of glycolipid-deficient compositions on the dynamic properties of the asymmetric MOM.

The asymmetric architecture of the mycobacterial outer membrane (MOM) establishes a vertical heterogeneity that underpins its unique biophysical and functional properties. The inner leaflet, composed of MAs, maintains a liquid ordered phase, while the outer leaflet, composed of loosely packed PDIMs and trehalose-derived glycolipids, retains a liquid disordered phase. This fluidity gradient diverges from conventional membrane systems, where disordered leaflets typically undergo induced ordering via interactions with adjacent ordered layers (*Feigenson et al., 2023*). In *Mtb*, however, PDIM and PAT disrupt interleaflet coupling, stabilizing vertical heterogeneity. Critically, this localized disorder in the outer leaflet may enable lipid shedding events, such as the release of immunomodulatory TDM. The heightened mobility of lipids in disordered domains could destabilize their membrane anchorage, allowing dissociation driven by mechanical instability and thermal fluctuations. While direct observation of shedding remains beyond current simulation resolution, this mechanism could reconcile *Mtb's* impermeability with its ability to dynamically interact with host immune systems, that is shedding lipids to modulate immunity while retaining barrier integrity. This vertical fluidity gradient contrasts with eukaryotic plasma membranes, which sustain functional asymmetry through cholesterol enrichment in the outer leaflet to stabilize microdomains (*Doktorova et al., 2025*; *Ikeda et al., 2006*). Unlike cholesterol-dependent eukaryotes, *Mtb* achieves its asymmetry through MA rigidity and outer leaflet lipid-driven disorder, bypassing sterols. The resulting 'rigid yet adaptive' architecture balances global impermeability with localized plasticity, permitting spatially restricted lipid rearrangements, which may aid in pathogenicity by TDM shedding at host interaction. Such compartmentalization would allow *Mtb* to fine-tune immune evasion while resisting small-molecule penetration.

The MOM's design further diverges from Gram-negative bacterial outer membranes, which rely on lipopolysaccharide (LPS)-stiffened outer leaflets and porin-mediated transport (*Nikaido, 2003*; *Silhavy et al., 2010*). While Gram-negative bacteria employ porins for selective permeability and efflux pumps for antibiotic resistance, *Mtb* lacks porins and instead relies on its densely packed, interdigitated MA layer to limit passive diffusion. Recent studies propose that PE/PPE proteins may act as selective transporters, suggesting a need for updated models integrating protein-lipid cooperativity (*Babu Sait et al., 2022*; *Boradia et al., 2022*). Moreover, the MOM's unique thermal resilience, evidenced by its ~338 K phase transition, may stem from cyclopropane modifications in α-MA chains that modulate fluidity without sacrificing structural coherence. This contrasts with Gram-negative bacteria's outer membranes, where LPS rigidity in the outer leaflet and phospholipid fluidity in the inner leaflet are spatially segregated between leaflets (*Im and Khalid, 2020*; *Wu et al., 2014*). *Mtb's* vertical heterogeneity thus represents an evolutionary innovation, enabling simultaneous thermal adaptation and pathogenicity.

Future studies should prioritize elucidating lipid trafficking mechanisms across the MOM's asymmetric leaflets, particularly how PDIMs and other lipids transit from outer to inner leaflets or vice versa, despite MA interdigitation. Computational models incorporating PE/PPE proteins could reveal dynamic protein-lipid interactions governing selective transport or shedding. Additionally, coarse-grained

models of the outer membrane could aid in drug-transport studies, potentially revealing energetic pathways by which novel antibiotics penetrate the complex cell envelope over larger timescales. Therapeutically, disrupting lipid clustering at the interleaflet interface or inhibiting enzymes responsible for MA cyclopropanation could destabilize the fluidity gradient, rendering *Mtb* vulnerable to antibiotic penetration while impairing immune evasion. Additionally, outer membranes of other mycobacteria can be modeled using the framework described here. For instance, adding unsaturated lipids and trehalose polyphleates while omitting PAT would allow researchers to investigate the unique properties of *Mycobacterium abscessus*. By bridging molecular-scale asymmetry with organism-scale resilience, this model advances our understanding of how *Mtb's* mycomembrane achieves extraordinary adaptability.

## Methods

### System and simulation parameters

All bilayer systems were constructed using *CHARMM-GUI Membrane Builder* (*Feng et al., 2023*; *Jo et al., 2009*) and were simulated using *OpenMM* version 8.2 under periodic boundary conditions (*Eastman et al., 2024*). The inner leaflet symmetric bilayers comprised α-MAs with 78 carbons, initialized in extended (eU), semi-folded (sZ), or fully folded (W) conformations (*Figure 2B–D*, *Supplementary file 1*). Outer leaflet symmetric bilayers included six *Mtb*-specific glycolipids: PDIM, TDM, TMM, DAT, PAT, and SGL (*Figure 2E–J*, *Supplementary file 1*), with compositions validated against experimental ratios (*Chiaradia et al., 2017*). Asymmetric bilayers combined MA-rich inner leaflets (94 eU, 47 sZ, 47 W MAs) and glycolipid-rich outer leaflets (20 lipids per type). Each system was solvated in a 150 mM KCl solution using the TIP3P water model (*Jorgensen et al., 1983*) and neutralized with counterions. Force field parameters for MAs and glycolipids were derived from the *CHARMM36* lipid force field (*Klauda et al., 2010*), with cyclopropane ring parameters validated against prior simulations (*Poger and Mark, 2015*). Following *Membrane Builder*'s default six-step equilibration protocol (*Jo et al., 2009*; *Jo et al., 2007*), the NVT (constant particle number, volume, and temperature) dynamics was first applied with a 1 femtosecond (fs) time step for 250 picoseconds (ps). Subsequently, the NPT (constant particle number, pressure, and temperature) ensemble was applied with a 1 fs time step (for 125 ps) and with a 2 fs time step (for 1.5 ns). During the equilibration, positional and dihedral restraint potentials were applied to lipid and water molecules, and their force constants were gradually reduced. With no restraints, we performed 3 μs production for three replicas of each system using a 4 fs timestep. Hydrogen mass repartitioning method was used together with the SHAKE algorithm to constrain the bonds containing hydrogen atoms (*Ryckaert et al., 1977*). The van der Waals interactions were cutoff at 12 Å with a force-switching function between 10 and 12 Å (*Steinbach and Brooks, 1994*), and electrostatic interactions were calculated by the particle-mesh Ewald method (*Essmann et al., 1995*). The temperature and pressure (at 1 bar) were controlled by Langevin dynamics with a friction coefficient of 1 ps$^{-1}$ and a semi-isotropic Monte Carlo barostat, respectively (*Åqvist et al., 2004*; *Chow and Ferguson, 1995*).

### Bilayer visualizations

Structural snapshots were rendered using *VMD 1.9.4* (*Humphrey et al., 1996*). Lipids were visualized with the *QuickSurf* drawing method to emphasize packing density, and oxygen atoms were highlighted as van der Waals spheres. For alignment of α-MA conformations (*Figure 3A, D and G*), C10–C27 carbons of the β-hydroxy chain were used as reference points. Lipid colors matched initial configurations (*Figure 2*) to ensure consistency across the figures.

### Bilayer analysis

*System width:* The system width of the simulation box was defined as the range of x values of the system at each time step. The average values in *Supplementary file 1* are an average over the final 500 ns of each replica.

   *Membrane thickness:* The hydrophobic membrane thickness was measured as the distance between the average z-positions of the first carbon in each lipid tail (C24 and C27 in MA-containing lipids; C1 in other lipids) from opposing leaflets.

*APL (area per lipid):* The APL was calculated by dividing the system's XY area at each time step by the number of lipids in each leaflet. The values in *Supplementary file 1* are an average over the final 500 ns of each replica.

*Z-density profiles:* The mass density profiles along the bilayer normal (z-axis) were generated using an in-house python script, with a 1 Å resolution. Terminal carbons (C75 of MAs), cyclopropane rings, and glycolipid headgroups were resolved separately.

*Scd vs Z profiles:* Deuterium order parameters ($S_{CD}$) were computed using an in-house python script with values averaged over the final 500 ns. $S_{CD}$ was calculated for all acyl carbons in the tails of each lipid type and binned into 2 Å slabs along the z-axis, using the following equation.

$$S_{CD} = \frac{\langle 3cos^2\theta\text{-}1 \rangle}{2}$$

where, $\theta$ is the angle between the C-H bond vector and the bilayer normal. The angular brackets represent temporal and molecular ensemble averages.

Lateral Diffusion Coefficients: The lateral diffusion coefficients ($D$) were calculated from the mean squared displacement (MSD) using the Einstein relation:

$$D = \frac{1}{4} \lim_{t \to \infty} \frac{d}{dt} \left\langle \left| r\left(t\right) - r\left(0\right) \right|^2 \right\rangle$$

MSD calculations were performed using MDAnalysis for the final 100 ns of simulation with a 10 ns lag time. The z-component of position was excluded to isolate lateral motions.

## Acknowledgements

This work is supported by the CNRS-MITI grant "Modélisation du vivant" 2020 (to MC), NSF MCB-2111728, and NIH R35 GM153458 (to WI). We thank Dr. Seonghoon Kim and Emanuel Luna for creating the topology and parameter files of the lipids in this study. We also thank Dr. Vivek Thacker and Dr. Gregor Weiss for invaluable discussions on the biological relevance of our simulations.

## Additional information

### Funding

| Funder | Grant reference number | Author |
|---|---|---|
| National Science Foundation | MCB-2111728 | Wonpil Im |
| National Institutes of Health | GM153458 | Wonpil Im |
| CNRS-MITI | "Modélisation du vivant" 2020 | Matthieu Chavent |

The funders had no role in study design, data collection and interpretation, or the decision to submit the work for publication.

### Author contributions

Turner P Brown, Formal analysis, Validation, Investigation, Visualization, Writing – original draft, Writing – review and editing; Matthieu Chavent, Conceptualization, Supervision, Validation, Writing – review and editing; Wonpil Im, Conceptualization, Supervision, Funding acquisition, Validation, Writing – review and editing

### Author ORCIDs

Turner P Brown ⬤ https://orcid.org/0009-0008-5100-629X
Matthieu Chavent ⬤ https://orcid.org/0000-0003-4524-4773
Wonpil Im ⬤ https://orcid.org/0000-0001-5642-6041

Reviewer #1 (Public review): https://doi.org/10.7554/eLife.108644.3.sa1
Reviewer #2 (Public review): https://doi.org/10.7554/eLife.108644.3.sa2
Author response https://doi.org/10.7554/eLife.108644.3.sa3

## Additional files

### Supplementary files

Supplementary file 1. Composition and structural properties of simulated membrane systems. System names indicate lipid composition and simulation temperature (K). Lipid composition is given as the number of molecules per leaflet in the order: MA_eU, MA_sZ, MA_W, PDIM, TDM, TMM, DAT, PAT, and SGL. Systems labeled 'both' are symmetric bilayers; asymmetric systems (*Asym_313* and *Asym_333*) are reported with inner and outer leaflets listed separately. System size corresponds to the lateral box dimension (Å), membrane thickness to the bilayer thickness (Å), and APL to the area per lipid (Å²). Reported structural properties for asymmetric systems correspond to the full bilayer.

MDAR checklist

### Data availability

The input and restart files necessary for the continuation of all simulations are freely available in a Zenodo dataset (https://doi.org/10.5281/zenodo.18246188). An example bash script for automatic job submission on HPCs, and detailed descriptions of included files can also be found in this dataset.

The following dataset was generated:

| Author(s) | Year | Dataset title | Dataset URL | Database and Identifier |
|---|---|---|---|---|
| Wonpil I, Matthieu C | 2026 | Dynamic Architecture of Mycobacterial Outer Membranes Revealed by All-Atom Simulations | https://doi.org/10.5281/zenodo.18246188 | Zenodo, 10.5281/zenodo.18246188 |

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
