## [Editor Report · eLife Assessment]

In their study, Brown et. al. provide an **important** advance in understanding the architecture of the mycobacterial outer membrane. Using all-atom simulations of model mycomembranes, the work reports **compelling** structural insights into how α-mycolic acids and outer leaflet lipids (PDIM and PAT) shape membrane organisation. The work revealed membrane heterogeneity with ordered inner leaflets and disordered outer leaflets that provide a molecular explanation for the resilience of the mycobacterial envelope.

---

## [Referee Report · Reviewer #1 (Public review)]

Disclaimer:

This reviewer is not an expert on MD simulations but has a basic understanding of the findings reported and is well-versed with mycobacterial lipids.

Summary:

In this manuscript titled "Dynamic Architecture of Mycobacterial Outer Membranes Revealed by All-Atom 1 Simulations", Brown et al describe outcomes of all-atom simulation of a model outer membrane of mycobacteria. This compelling study provided three key insights:

(1) The likely conformation of the unusually long chain alpha-branched, beta-methoxy fatty acids-mycolic acids in the mycomembrane to be the extended U or Z type rather than the compacted W-type.

(2) Outer leaflet lipids such as PDIM and PAT provide regional vertical heterogeneity and disorder in the mycomembrane that is otherwise prevented in a mycolic acid only bilayer.

(3) Removal of specific lipid classes from the symmetric membrane systems lead to significant changes in membrane thickness and resilience to high temperatures. (4) The asymmetric mycomembrane presents a phase transition from a disordered outer leaflet to an ordered inner leaflet.

Strengths:

The authors take a stepwise approach to increasing the membrane's complexity and highlight the limitations of each approach. A case in point is the use of supraphysiological temperatures of 333 K or higher in some simulations. Overall, this is a very important piece of work for the mycobacterial field and will likely help develop membrane-disrupting small molecules and provide important insights into lipid-lipid interactions in the mycomembrane.

Weaknesses:

The authors used alpha-mycolic acids only for their models. The ratios of alpha-, keto-, and methoxy-mycolic acids are well documented in the literature, and it may be worth including them in their model. Future studies can aim to address changes in the dynamic behavior of the MOM by altering this ratio, but including all three forms in the current model will be important and may alter the other major findings of the current study.

---

## [Referee Report · Reviewer #2 (Public review)]

Summary:

The manuscript reports all-atom molecular dynamics simulations on outer membrane of *Mycobacterium tuberculosis*. This is the first all-atom MD simulation of MTb outer membrane and complements the earlier studies which used coarse-grained simulation.

Strengths:

The simulation of outer membrane consisting of heterogeneous lipids is a challenging task and the current work is technically very sound.

The observation about membrane heterogeneity and ordered inner leaflets vs disordered outer leaflets is a novel result from the study. This work will also facilitate other groups to work on all atom models of mycobacterial outer membrane for drug transport etc.

Comments on revisions:

I would like to thank the authors for addressing all the concerns and providing additional details to improve the clarity of presentation.

---

## [Author Response]

The following is the authors’ response to the original reviews.

**Public Reviews:**

**Reviewer #1 (Public review):**
Summary:In this manuscript titled "Dynamic Architecture of Mycobacterial Outer Membranes Revealed by All-Atom 1 Simulations", Brown et al describe outcomes of all-atom simulation of a model outer membrane of mycobacteria. This compelling study provided three key insights:(1) The likely conformation of the unusually long chain alpha-branched beta-methoxy fatty acids, mycolic acids in the mycomembrane, to be the extended U or Z type rather than the compacted W-type. (2) Outer leaflet lipids such as PDIM and PAT provide regional vertical heterogeneity and disorder in the mycomembrane that is otherwise prevented in a mycolic acid-only bilayer. (3) Removal of specific lipid classes from the symmetric membrane systems leads to significant changes in membrane thickness and resilience to high temperatures.

In addition to the three key insights, we would like to add one more; (4) asymmetric mycomembrane presents a phase transition from a disordered outer leaflet to an ordered inner leaflet.

Strengths:The authors take a step-wise approach in building the complexity of the membrane and highlight the limitations of each of the approaches. A case in point is the use of supraphysiological temperature of 333 K or even higher temperatures for some of the simulations. Overall, this is a very important piece of work for the mycobacterial field, and will help in the development of membrane-disrupting small molecules and provide important insights for lipid-lipid interactions in the mycomembrane.

We appreciate Reviewer’s positive view on our work.

Weaknesses:(1) The authors used alpha-mycolic acids only for their models. The ratios of alpha, keto, and methoxy-mycolic acids are known in the literature, and it may be worth including these in their model. Future studies can be aimed at addressing changes in the dynamic behavior of the MOM by altering this ratio, but the inclusion of all three forms in the current model will be important and may alter the other major findings of the current study.

We agree that adjusting the ratios of mycolates may impact the dynamic behavior of the MOM. However, including various ratios of these lipids would require much work and introduce unnecessary complexity to our model; believe or not, the current work took more than 3 years. Investigations into the effects of mycolate structure in the MOM would be interesting and suitable for future studies.

(2) The findings from the 14 different symmetric membrane systems developed with the removal of one complex lipid at a time are very interesting but have not been analysed/discussed at length in the current manuscript. I find many interesting insights from Figures S3 and S5, which I find missing in the manuscript. These are as follows:(a) Loss of PDIM resulted in reduced membrane thickness. This is a very important finding given that loss of PDIM can be a spontaneous phenomenon in Mtb cultures in vitro and that this is driven by increased nutrient uptake by PDIM-deficient bacilli (Domenech and Reed, 2009 Microbiology). While the latter is explained by the enhanced solute uptake by several PE/PPE transporter systems in the absence of PDIM (Wang et al, Science 2020), the findings presented by Brown et al could be very important in this context. A discussion on these aspects would be beneficial for the mycobacterial community.

Following Reviewer’s suggestion, we have added the following to the Discussion section.

“The outer leaflet symmetric bilayers, comprised of trehalose-derived glycolipids and PDIMs, reveal PDIM-dependent thickness. As observed in both symmetric outer leaflet systems and asymmetric systems, PDIM migrates to the bilayer midplane, causing the upper leaflet to bulge and increasing the overall thickness. Reduced thickness in the systems lacking PDIM, an important virulence factor for *Mtb*, may allow for higher nutrient uptake. This corroborates a 2009 study in which Domenech and Reed found a correlation between PDIM absence in vitro and attenuated virulence (Domenech and Reed, 2009).”

(b) I find it interesting that loss of PAT or DAT does not change membrane thickness (Figure S3). While both PAT and PDIM can migrate to the interleaflet space, loss of PDIM and PAT has a different impact on membrane thickness. It is worth explaining what the likely interactions are that shape membrane thickness in the case of the modelled MOM.

We have added the following to the section titled “Outer leaflet lipids drive unexpected membrane heterogeneity and softness of the Mycomembrane”.

“Although PAT also migrates to the bilayer midplane, the PAT-deficient bilayers did not exhibit reduced thickness as the PDIM-deficient thickness did (Supporting Information Table S1). This may be due to fewer PAT than PDIM moving to the bilayer midplane. In the All_Lipids systems, PDIM migrates first, bulging the upper leaflet and reducing lipid headgroup crowding (Supporting Information Figs. S5, S6). In this slightly less crowded environment, hydrophobic forces from PAT’s tails overcome the hydrophilic forces from the trehalose headgroup, causing some PATs to move deeper into the hydrophobic region.”

(c) Figure S5: Is the presence of SGL driving PDIM and PAT to migrate to the inter-leaflet space? Again, a discussion on major lipid-lipid interactions driving these lipid migrations across the membrane thickness would be useful.

We have added the following to the section titled “Outer leaflet lipids drive unexpected membrane heterogeneity and softness of the Mycomembrane”.

“Additionally, in SGL-deficient bilayers, fewer PDIMs and PATs move to the bilayer midplane. This may be due to the highly methylated lipid tails of SGL. When present in the bilayer, these methyl groups may disrupt lipid packing and increase fluidity, allowing more PDIMs to move into the hydrophobic region. Supporting Information Figure S8 shows the average lipid order parameter along each lipid tail for all outer leaflet symmetric systems. Without SGL, lipid tails are consistently more ordered, supporting the notion that SGL’s methylated tails are disrupting lipid packing. Further studies are necessary to investigate the effect of glycolipid-deficient compositions on the dynamic properties of the asymmetric MOM.”

**Reviewer #2 (Public review):**
Summary:The manuscript reports all-atom molecular dynamics simulations on the outer membrane of *Mycobacterium tuberculosis*. This is the first all-atom MD simulation of the MTb outer membrane and complements the earlier studies, which used coarse-grained simulation.

The Reviewer is correct in that this is the first MD simulation of the Mtb outer membrane with diverse lipid types.

Strengths:The simulation of the outer membrane consisting of heterogeneous lipids is a challenging task, and the current work is technically very sound. The observation about membrane heterogeneity and ordered inner leaflets vs disordered outer leaflets is a novel result from the study. This work will also facilitate other groups to work on all-atom models of mycobacterial outer membrane for drug transport, etc.

We appreciate Reviewer’s positive view on our work.

Weaknesses:Beyond a challenging simulation study, the current manuscript only provides qualitative explanations on the unusual membrane structure of MTb and does not demonstrate any practical utility of the all-atom membrane simulation. It will be difficult for the general biology community to appreciate the significance of the work, based on the manuscript in its current form, because of the high content of technical details and limited evidence on the utility of the work.Major Points:(1) The simulation by Basu et al (Phys Chem Chem Phys 2024) has studied drug transports through mycolic acid monolayers. Since the authors of the current study have all atom models of MTb outer membrane, they should carry out drug transport simulations and compare them to the outer membranes of other bacteria through which drugs can permeate. In the current manuscript, it is only discussed in lines 388-392. Can the disruption of MA cyclopropanation be simulated to show its effect on membrane structure?

We acknowledge the potential for simulations of drug transport through our MOM model. However, we believe with the current timescale, these simulations may be better suited for a coarse-grained model of the MOM. We plan to do this in the future, but it is out of the scope of the current study. We have added the following to the Discussion section to address this point.

“Additionally, coarse-grained models of the outer membrane could aid in drug-transport studies, potentially revealing energetic pathways by which novel antibiotics penetrate the complex cell envelope over larger timescales.”

(2) In line 277, the authors mention about 6 simulations which mimic lipid knockout strains. The results of these simulations, specifically the outcomes of in silico knockout of lipids, are not described in detail.

We have added the following to the Discussion section to show the effect of glycolipid composition on the deuterium order parameter.

“The outer leaflet symmetric bilayers, comprised of trehalose-derived glycolipids and PDIMs, reveal PDIM-dependent thickness. As observed in both symmetric outer leaflet systems and asymmetric systems, PDIM migrates to the bilayer midplane, causing the upper leaflet to bulge and increasing the overall thickness. Reduced thickness in the systems lacking PDIM, an important virulence factor for *Mtb*, may allow for higher nutrient uptake. This corroborates a 2009 study in which Domenech and Reed found a correlation between PDIM absence in vitro and attenuated virulence (Domenech and Reed, 2009). Although PAT also migrates to the bilayer midplane, the PAT-deficient bilayers did not exhibit reduced thickness as the PDIM-deficient thickness did. This may be due to fewer PAT than PDIM moving to the bilayer midplane. In the All_Lipids systems, PDIM migrates first, bulging the upper leaflet and reducing lipid headgroup crowding. In this slightly less crowded environment, hydrophobic forces from PAT’s tails overcome the hydrophilic forces from the trehalose headgroup, causing some PATs to move deeper into the hydrophobic region. Additionally, in SGL-deficient bilayers, fewer PDIMs and PATs move to the bilayer midplane. This may be due to the highly methylated lipid tails of SGL. When present in the bilayer, these methyl groups may disrupt lipid packing and increase fluidity, allowing more PDIMs to move into the hydrophobic region. Supporting Information Figure S8 shows the average lipid order parameter along each lipid tail for all outer leaflet symmetric systems. Without SGL, lipid tails are consistently more ordered, supporting the notion that SGL’s methylated tails are disrupting lipid packing. Further studies are necessary to investigate the effect of glycolipid-deficient compositions on the dynamic properties of the asymmetric MOM.”

(3) Figure 5 shows PDIM and PAT-driven lipid redistribution, which is a significant novel observation from the study. However, comparison of 3B and 3D shows that at 313K, the movement of the PDIM head group is much less. Since MD simulations are sensitive to random initial seeds, repeated simulations with different random seeds and initial structures may be necessary.

The difference in headgroup movement at different temperatures can be attributed to higher kinetics at 333K, causing the lipids to move faster. The relatively slow speed and computational load of running all-atom simulations make it difficult to simulate these lower temperatures on the timescales necessary to observe full aggregation of PDIM. However, CG simulations may be sufficient to sample these events. We have addressed this by adding the following to the Results section.

“We also observed a stark difference in the speed with which PDIM and PAT migrate to the center at different temperatures. PDIM molecules do not fully aggregate at the membrane center until about 1500 ns at 313K, whereas they accumulate within 500 ns at 333K (Fig. 5B, 5D). This can be attributed to higher kinetics at 333K, causing the lipids to move faster. Coarse-grained models may be sufficient to observe full aggregation of hydrophobic species at the membrane midplane at lower temperatures.”

(4) As per Figure 1, in the initial structure, the head group of PAT should be on the membrane surface, similar to TDM and TMM, while PDIM is placed towards the interior of the outer membrane. However, Figure 5 shows that at t=0, PAT has the same Z position as PDIM. It will be necessary to provide Z-position Figures for TMM and TDM to understand the difference. Is it really dependent on the chemical structure of the lipid moiety or the initial position of the lipid in the bilayer at the beginning of the simulation?

We have added the following to the Results section to address this comment.

“In all symmetric outer leaflet simulations, PDIM and PAT sit just below the headgroups of other lipids at the start of production, due to our equilibration scheme. During the last step of equilibration, lipid headgroups are allowed to move freely, which initiates migration to the membrane center and causes the slight difference between PDIM/PAT and the other lipids’ headgroup positions (Supporting Information Figs. S5, S6).”

Minor Point:In view of the complexity of the system undertaken for the study, the manuscript in its current form may not be informative for readers who are not experts in molecular simulations.

This work represents the first atomistic simulation of the mycobacterial outer membrane. While not perfectly realistic, as it does not include arabinogalactan or peptidoglycan, it does have extensive descriptions of each lipid simulated and their relevance to the survival of *Mtb*.

**Recommendations for the authors:**

**Reviewer #2 (Recommendations for the authors):**
(1) The interface to build and set up all atom coordinates of the outer membrane of *Mycobacterium tuberculosis* should be available from CHARMM-GUI.

The current manuscript is meant as a proof of concept for simulating bilayers composed of complex mycobacterial lipids. The current study itself took more than 3 years. Since we have developed CHARMM-GUI, the lipids described in this paper may be available in CHARMM-GUI in the future, but that is not the aim of this paper. Initial structures and final 50 ns of the simulations are available to readers (see Data Acknowledgements).

(2) The difference between symmetric and asymmetric systems in Figures 2K and 2L is not at all clear, neither in the legend to the figure nor in the manuscript text. The color codes in 2K and 2L should be described with clarity. The authors should provide schematic diagrams similar to Figure 1 to explain each of the simulation systems they are discussing. This will clarify the difference between symmetric and asymmetric systems.

We have updated Figure 1 to clearly show which systems are symmetric and which are asymmetric.

(3) The first two sub-sections of the RESULT section discuss symmetric mycolic acid bilayers. The observations on thermal resilience and phase transitions are interesting, but the relevance of symmetric mycolic acid bilayers (Figures 3 & 4) to the major focus of the current manuscript (i.e., outer membrane consisting of multiple lipids) is not clear.

Most previous simulations only focused on monolayers of mycolic acids. Our symmetric bilayers are used to provide reasonable APL and system compositions for the asymmetric membrane, so as to avoid area mismatch. We can also gain insights into how these unique lipids behave in symmetric bilayers, which may be useful to scientists aiming to study simpler membranes in the context of drug permeation or pore formation. These points have been addressed in the following addition to the Introduction section.

“We have also used the equilibrated symmetric bilayers to estimate reasonable areas per lipid and facilitate the modeling of stable asymmetric systems.”